# Isolation of High Molecular Weight DNA from the Model Beetle *Tribolium* for Nanopore Sequencing

**DOI:** 10.3390/genes12081114

**Published:** 2021-07-22

**Authors:** Marin Volarić, Damira Veseljak, Brankica Mravinac, Nevenka Meštrović, Evelin Despot-Slade

**Affiliations:** Division of Molecular Biology, Ruđer Bošković Institute, Bijenička cesta 54, 10000 Zagreb, Croatia; marin.volaric@irb.hr (M.V.); damira.veseljak@irb.hr (D.V.); brankica.mravinac@irb.hr (B.M.); nevenka@irb.hr (N.M.)

**Keywords:** beetle, high molecular weight DNA, Nanopore sequencing

## Abstract

The long-read Nanopore sequencing has been recently applied for assembly of complex genomes and analysis of linear genome organization. The most critical factor for successful long-read sequencing is extraction of high molecular weight (HMW) DNA of sufficient purity and quantity. The challenges associated with input DNA quality are further amplified when working with extremely small insects with hard exoskeletons. Here, we optimized the isolation of HMW DNA from the model beetle *Tribolium* and tested for use in Nanopore sequencing. We succeeded in overcoming all the difficulties in HMW handling and library preparation that were encountered when using published protocols and commercial kits. Isolation of nuclei and subsequent purification of DNA on an anion-exchange chromatography column resulted in genomic HMW DNA that was efficiently relaxed, of optimal quality and in sufficient quantity for Nanopore MinION sequencing. DNA shearing increased average N50 read values up to 26 kb and allowed us to use a single flow cell in multiple library loads for a total output of more than 13 Gb. Although our focus was on *T. castaneum* and closely related species, we expect that this protocol, with appropriate modifications, could be extended to other insects, particularly beetles.

## 1. Introduction

The beetle *Tribolium castaneum* has become one of the most important models in the field of evolution, physiology and development because its development is more representative for insects compared to Drosophila [1]. It is a worldwide pest of stored products and represents the most species-rich animal order on Earth, the coleopterans. *T. castaneum* genome has been sequenced, annotated and a reference genome is available [2,3]. The estimated genome size of 204 Mb is 44 Mb larger than the assembled genome sequence [4] suggesting that almost 25% of the genome remains unassembled. It is not unexpected taking into account that structural genome analyses revealed large amounts of different classes of repetitive DNA especially in abundant (peri) centromeric regions [2,5] which are therefore underrepresented or even absent in the genome assemblies [6] which results in assembly gaps. An optimistic perspective for analysis of linear genome organization of *T. castaneum* genome using the Nanopore approach is found in a recently published high-quality assembly of a human genome where Nanopore sequencing strategy spans hundreds of kilobases of highly repetitive DNA [7,8,9]. Moreover, Nanopore sequencing enables to directly detect different methylation states of bases in DNA [10,11], thus opening possibilities to explore *Tribolium* epigenome from a single sequencing run.

Nanopore sequencing by ONT surpasses potentially unlimited read length and a decent sequencing read accuracy (>95%) as it is based on novel principle of detecting the change in ionic current as a single-stranded DNA passes through a protein nanopore embedded in an electrically resistant polymer membrane. Because of these unique properties, one of the main factors for successful sequencing is the quality of the starting DNA. There are three main criteria that isolated genomic DNA should meet. First, it must be devoid of impurities and contaminants, which can be measured with a spectrophotometer and evaluated by absorbance ratios. Additionally, its concentration must be correctly quantified and quantifiably stable throughout measurements, since it is known that HMW DNA tends to be poorly soluble, which can be assisted by longer relaxation times. Finally, to achieve long reads in sequencing, the molecular weight of DNA should be in the desired range meeting the end goal, which is measured by pulsed field gel electrophoresis (PFGE) analysis.

With the development of Nanopore sequencing technologies, the need for optimized protocols for HMW DNA extraction suitable for library preparation has increased. There are a variety of methods describing the isolation of HMW DNA from cells and specific tissues [12,13]. Tissue isolation usually requires optimization, depending on the condition and availability of the starting material. Insects further represent a challenging material as they are the most diverse group of animals. Although some of DNA extractions have been evaluated for beetles [14] and are capable of producing DNA of moderately high molecular weight (~40 kb), none of them have been tested for use in Nanopore sequencing. In addition, it is crucial to have a method that allows modifications to tailor it to specific scientific questions needed to be answered. Several commercial kits have been developed to isolate HMW genomic DNA suitable for long read sequencing. These kits are based on one of the main principles for isolating nucleic acids: silica-gel based membranes, anion-exchange chromatography columns, magnetic beads and disks with innovative silica-based chemistry, a glass bead-based approach, or the salting-out technique. However, there are also more conventional methods such as grinding in liquid nitrogen and using phenol-chloroform or another extraction buffer. There is also a way to isolate and embed chromosome-sized HMW DNA from nuclei into agarose plugs and the protocol has recently been developed in insects for BAC library construction [15].

Here, we developed a HMW DNA extraction protocol that proved to be optimal for library preparation and Oxford Nanopore MinION sequencing using the beetle *T. castaneum* and closely related species as a model system.

## 2. Materials and Methods

### 2.1. Materials

Insects: Laboratory culture of the red flour beetle *Tribolium castaneum*; highly inbred Georgia 2 (GA2) strain, *T. freemani* and *T. confusum* routinely reared in whole wheat flour at 28 °C and 40% relative humidity in the dark. For the collection of a larger amount of different life stages, 0.71 mm sieve was used for sifting and individual beetles were picked with tweezers.Liquid nitrogen.Sterilized mortar with pestle and metal spatula.100 µm cell strainer (Thermo Fisher Scientific, Cat. No. 22363549, Waltham, MA, USA).1000 µL wide bore filtered pipette tips (Thermo Fisher Scientific, Cat. No. 2079G, Waltham, MA, USA).1.5 mL DNA LoBind tubes (Eppendorf, Cat. No. 0030108051, Hamburg, Germany).Glass rod (Pasteur pipette heated at the end to form thin hook).NIB buffer: 10 mM Tris pH 9.4, 60 mM NaCl, 10 mM EDTA pH 8.0, 0.15 mM spermidine, 0.15 mM spermine, 0.5% *v*/*v* Triton X-100, 0.1% *v*/*v* β-mercaptoethanol.Blood and Cell Culture DNA Midi Kit (Qiagen, Cat. No. 13343, Hilden, Germany) or Genomic-Tip 100/G columns (Qiagen, Cat. No. 10243) with prepared G2 (800 mM guanidine HCl, 30 mM Tris-Cl pH 8.0, 30 mM EDTA pH 8.0; 5% Tween-20, 0.5% Triton X-100), QBT (750 mM NaCl, 50 mM MOPS pH 7.0, 15% isopropanol, 0.15% triton X-100), QC (1 M NaCl, 50 mM MOPS pH 7.0, 15% isopropanol) and QF (1.25 M NaCl, 50 mM Tris-Cl pH 8.5, 15% isopropanol) buffers according to recipes of manufacturer’s kit handbook.RNase A solution (100 mg/mL, Qiagen, Cat. No. 1007885).Proteinase K from Tritirachium album (Sigma-Aldrich, Cat. No. SRE0047, St. Louis, MO, USA) or Protease (7.5 AU, Qiagen, Cat. No. 19155), prepared as 20 mg/mL solution in miliQ water.Isopropyl alcohol.TE buffer pH 8.0: 10 mM Tris, 1 mM EDTA.Nuclease-free water (Invitrogen, Cat. No. AM9937, Waltham, MA, USA).Pulsed Field Certified Agarose (BioRad, Cat. No. 1620137, Hercules, CA, USA).0.5× TBE buffer: 45 mM Tris-borate, 1 mM EDTA.Quick-Load 1 kb Extend DNA Ladder (New England Biolabs, Cat. No. N3239S, Ipswich, MA, USA).Lambda PFG Ladder (New England Biolabs, Cat.No. N0341S).Syringe and G30 needle.Short Read Eliminator XS Kit (Circulomics, Cat. No. SKU SS-100-121-01, Baltimore, MD, USA).Qubit dsDNA BR Assay kit (Invitrogen, Cat. No. Q32850).Agencourt AMPure XP beads (Beckman Coulter, Cat. No. A63880, Brea, CA, USA).Ligation Sequencing Kit (Oxford Nanopore Technologies, Cat. No. LSK-110, Oxford, UK).NEBNext Companion Module for Oxford Nanopore Technologies Ligation Sequencing (New England BioLabs, Cat. No. E7180S).MinION flow cell (Oxford Nanopore Technologies, Cat. No. FLO-MIN111).

### 2.2. Equipment

#### 2.2.1. Necessary Equipment

Fume hood.Centrifuge with cooling option (Centrifuge 5424 R, Eppendorf).Thermomixer (ThermoMixer C with SmartBlock 1.5 mL and ThermoTop, Eppendorf).Shaker (Vibramax 100, Heidolph, Schwabach, Germany).Qubit 4 Fluorometer (Invitrogen).Spectrophotometer (BioSpec-nano, Shimadzu, Kyoto, Japan).

#### 2.2.2. Optional Equipment (for DNA Length Assessment, Library Preparation and Sequencing)

Magnetic separator for 1.5 mL tubes (MagnaRack Magnetic Separation Rack, Invitrogen).Pulsed field gel electrophoresis system (CHEF-DR III system, Bio-Rad).Thermal cycler (2720 Thermal cycler, Thermo Fisher Scientific).Rotator mixer (Programmable Rotator Multi Bio RS-24, Biosan, Riga, Latvia).MinION device (Oxford Nanopore Technologies).

### 2.3. Procedure

#### 2.3.1. Nuclei Isolation

The nuclei were isolated according to the protocol of Brown and Coleman [15] with a few modifications. Main modifications include the mortar and spatula being precooled in liquid nitrogen rather than the −80 °C, preparation of fresh NIB buffer immediately before use, an additional washing step of the isolated nuclei, as well as modification of the centrifuge times and usage of standard plastic tubes.

Immediately before starting, 20 mL of fresh NIB buffer per reaction was prepared and chilled on ice. The mortar was filled twice with liquid nitrogen, in order to sufficiently precool both the mortar and the spatula. During the second nitrogen evaporation, the sample in amounts according to Table 1 and Appendix A was added and grounded in a circular motion. The grinding pressure and the speed were slowly increased in order to produce a fine powder. Using the spatula, all of the powder was scraped into a 50 mL tube containing 8 mL of chilled NIB. In order to achieve maximum efficiency the tube was gently swirled or, if residue remained on the tube walls a wide bore tip was used to gently flush the residue. It is important not to shake the tube. The suspension was transferred through a 100 µm cell strainer into a new chilled 50 mL tube.

The solution was divided into six chilled 1.5 mL tubes and spun at 100× *g* for 30 s at 4 °C. The supernatant was transferred into six new 1.5 mL tubes using a 1 mL wide bore tip carefully without disturbing the loosely adhering cell debris pellet. The tubes were once again placed in the centrifuge for 3 min at 1800× *g* and 4 °C to pellet the nuclei. The supernatant was discarded and as much liquid as possible was removed using a regular 1 mL tip. The pellet of compact nuclei was resuspended completely in 1 mL of cold NIB buffer by pipetting with a wide bore tip, but carefully as to avoid introducing air bubbles into the mixture. The previous centrifugation step was repeated. The lysis buffer was prepared by adding 500 µL Protease or 95 µL of Proteinase K and 10 µL of RNase A to 5 mL of G2 buffer. The supernatant after the last centrifugation step was removed and the pellet resuspended in 800 µL of the prepared G2 buffer. It is important to completely resuspend the pellet by pipetting with a wide bore tip, but again, carefully as to not introduce air bubbles. The tubes were incubated at 50 °C for 1 h at 300 rpm in a thermomixer. During incubation, it is recommended either to invert the tubes several times or gently pipette with wide bore tips in order to ensure complete digestion. The resuspended and properly digested nuclei had a milky, stringy texture and were visible to the naked eye. If hard clumps of nuclei remained, they could have been broken by additional pipetting.

#### 2.3.2. Genomic Tip Purification

Genomic Tip manipulation was performed according to the manufacturer’s protocol with few modifications, mainly the added pressure during all steps and mandatory prewarming of the QF buffer.

First, a Genomic Tip 100/G column was equilibrated with 4 mL of QBT buffer. After the digestion step, all of the prepared solutions were applied to the column. The column was washed twice with 7.5 mL of QC buffer. The final elution was performed with 5 mL of QF buffer prewarmed to 50 °C. For all steps, it is necessary to apply some pressure with a syringe plunger or a suitable rubber pipette to ensure sufficient flow rate.

The isolated DNA was precipitated by gently adding 3.5 mL of isopropanol at room temperature. The tube was left to stand for 30 s to allow the phases to separate completely. The upper phase should turn a whitish color. The tube was turned over at least 20 times and white DNA strands that stick together with each turn should appear in the solution. The white strands of DNA form a sticky DNA “jelly” which was then picked up with the tip of a thin glass rod. The DNA should stick to the glass rod and not fall off easily. The DNA was spooled and transferred to a 1.5 mL DNA LoBind tube in 100 µL elution buffer of choice (e.g., TE pH 8.0, 10 mM Tris-HCl pH 8.5). The DNA was incubated at 50 °C for up to 2 h to accelerate the homogenization process. At the end of the incubation, the DNA is usually almost completely dissolved, except for single filamentous clumps. The isolated DNA was left overnight at room temperature with gentle horizontal shaking (150 rpm) to achieve final relaxation. At the end, the DNA solution should be completely clear, with increased viscosity, which can be observed by flicking the tube. Subsequent storage was at 4 °C with no noticeable decrease in DNA quality and length for up to several months.

#### 2.3.3. DNA Shearing and Size Selection

The solution of homogenized DNA was sheared 10–30× with a 30-gauge needle. The concentration was then measured in triplicates and adjusted to 150 ng/µL with TE buffer or water. For size selection, the Short Read Eliminator (SRE) XS kit was used according to the manufacturer’s protocol. The final resuspension was done in 50 µL EB buffer from SRE kit and the concentration was measured twice to check concentration measurement reproducibility.

#### 2.3.4. Assessment of DNA Quality and Length

DNA concentration was always tested by fluorometric and spectrophotometric methods. Quality was checked with a spectrophotometer and acceptable values are: ~1.8 for A260/280 and ~2.2 for A260/230, according to the official recommendations of ONT. Pulsed field gel electrophoresis was used to assess the length of isolated DNA, sheared DNA and prepared library. DNA fragments were separated by PFGE on a 1% agarose gel run in 0.5× TBE buffer at 6 V/cm, 14 °C, included angle of 120°, switch time 1–10 s for 14 h using a Bio-Rad CHEF-DR III PFGE system. The gel was then stained in 1µg/mL ethidium bromide solution at RT on a shaker for 30 min.

#### 2.3.5. Nanopore Sequencing

For library preparation, 3–3.5 µg of DNA was used for a single reaction and all incubation times were extended as suggested in the Beads free library preparation protocol [16]. A MinION flow cell was loaded with 400–600 ng of DNA and run for 4–10 h before pausing and washing the cell. A prepared library was usually divided into two to three runs with up to five successful runs on one flow cell.

## 3. Results

To optimize the DNA extraction procedure, we used *T. castaneum* from different developmental stages (larvae, pupae, adults) together with two other *Tribolium* species (*T. freemani, T. confusum*). We first tested the E.Z.N.A kit (Omega BioTek, Norcross, GA, USA), the Monarch HMW DNA extraction kit for tissue (New England Biolabs), the Blood and Cell Culture DNA Mini and Midi kit (Qiagen) and phenol-chloroform extraction. Observed strengths and limitations of these approaches are elaborated in detail in Appendix A. In summary, limitation for the use of DNA obtained with all tested commercial kits and phenol-chloroform extraction in library preparation and manipulation for Nanopore sequencing is primarily due to insoluble DNA pellets that also had impurities, as evident from poor absorbance ratios. Subsequently, when DNA introduced into the standard library preparation protocols of ONT, it resulted in clumping of magnetic beads, large losses after each step of library preparation. Even though some kits were able to produce DNA of considerable length, the quality of subsequent libraries was poor, with insufficient read lengths in Nanopore sequencing and rapid pore death (Appendix A). Therefore, we developed a HMW extraction protocol from cell nuclei with a purification step using commercially available Genomic Tip columns followed by DNA shearing and size selection (Figure 1).

The developed procedure, in combination with the slightly modified standard library preparation protocol (described in Materials and Methods section), showed the highest degree of reproducibility, optimal HMW DNA for Nanopore sequencing and always yielded DNA of sufficient quality and quantity for multiple rounds of sequencing. Results of Nanopore sequencing performed on HMW DNA isolated by using three different approaches are depicted on Figure 2. The last two plots have the highest N50 values and were obtained from DNA isolated from nuclei and purified on Genomic-Tip.

The optimized procedure was tested on all *T. castaneum* developmental stages (larvae, pupae, adults) and the DNA obtained had absorption ratios in the proposed range for sequencing (Table 1). DNA from pupae was isolated from approximately 200 mg of tissue and the obtained DNA yield was sufficient and in concentration adequate for size selection and library preparation. In adults and larvae, in order to achieve sufficient amount of DNA, more than 600 mg of starting tissue is required. This is due to the large amount of non-cellular material found in these developmental stages, mainly chitin in the form of the beetle cuticle in adults and vast amounts of fat and gut in the larvae.

The HMW DNA size distribution was examined by PFGE. The extracted DNA from *T. castaneum* showed most gDNA fragments distributed between 50 and 150 kb (Figure 3a). The DNA isolated from the pupal stage even showed an additional band at 200 kb. In addition, the same HMW DNA isolation procedure was tested for two congeneric species, *T. confusum* and *T. freemani*, which also had yielded gDNA up to 100 kb in length (Figure 3b).

In order to obtain more efficient sequencing, library preparations from sheared DNA were tested. Both pupal and larval DNA after shearing showed a decrease in the ultra-long DNA fraction with a majority of DNA in the 30–80 kb range. This is further tested with shear intensity assessment, where 30 passes through the G30 needle yielded the most compact band, with the majority still above 48 kb (Figure 3c). Interestingly and of great importance for subsequent sequencing, there was no clear increase in the abundance of shorter fragments in gel electrophoresis. Size selection on sheared DNA showed only a slight negative effect on DNA length in PFGE, attributed to an additional step of centrifugation and manipulation, which is indicated by a slight downward shift. Nonetheless, this step resulted in a reduction of short reads during sequencing, leading to longer pore lifetimes and greater overall yield. Because ultra-long DNA in library preparation did not increase N50 in sequencing, we decided to perform DNA needle shearing, which was shown to be beneficial [17]. Interestingly, we were able to increase N50 from about 11 kb to 20 kb after 20 passes through the G30 needle, and up to 26 kb when 30 shear passes were performed (Figure 4a–c).

To remove smaller DNA fragments, we used centrifugation-based size selection, which efficiently removes the majority of reads below 10 kb, which is not apparently visible on the PFGE but rather makes an immense difference in the sequencing run. This is best seen in the length histogram by the absence of the leftmost shortest read peak (Figure 4d).

Nanopore sequencing of the prepared library on the MinION cell with five consecutive loads within 48 h yielded 13.17 Gb of data (Figure 5a). The distribution of the Phred quality score (Q) shows that most reads have a quality above Q20, which means less than 1% error rate (Figure 5b). Moreover, quality correlates positively with read length, with the majority of longest reads having 99.6% accuracy (Q24).

## 4. Discussion

Following procedures for commercial kits, especially those made for smaller inputs and even tested for *Tribolium castaneum* [18] as well as phenol-chloroform protocol, did not yield DNA of sufficient quality and quantity for Nanopore sequencing. The Brown and Coleman’s protocol [15] proved to be the best starting point; however, although claimed to be suitable for long-range sequencing and optical mapping, this protocol does not elaborate on how to isolate HMW from plugs. Although it is likely to yield DNA of higher molecular weight, there is no information on exact lengths and qualities. Here, we have succeeded in isolating DNA and validating its use for library preparation and Nanopore sequencing.

All commercial kits tested provided DNA of sufficient length but had one major drawback; the DNA pellets were compact and difficult to dissolve posing a major problem during the various DNA clean-up and centrifugation steps required in library preparation. This difficulty was successfully overcome by using liquid nitrogen pulverization as the initial step, as this resulted in superior DNA quality, yield and reproducibility.

We found that the amount of input material is critical and, for beetles, depends on which developmental stage is used. For example, larvae and adults require more starting material due to the high proportion of non-cellular material (fat and chitin). This is most likely the main reason for the nonlinear dependence of input weight and final DNA concentration.

In the method we described, there are some steps that we think need special attention and can lead to unsuccessful isolation if not performed properly. Nuclear pellets should be completely resuspended before incubation with protease and RNase A. During subsequent incubation, additional pipetting can further aid in homogenization of the solution. The use of increased input material almost always resulted in poor flow rates in the purification step of the Genomic Tip, but had no negative effect on the final DNA and was resolved by the use of significant positive pressure using a propipette or syringe. A great indicator of the higher DNA concentrations and likely length, is the viscosity of the DNA eluting from the columns as falling droplets are left behind and expand. When precipitating DNA with isopropanol, it is advisable to keep the two phases separate before inverting so that the upper phase becomes whitish, which facilitates the formation of DNA fibers. Sometimes up to 30 inversions are necessary for compact DNA aggregates to form. We have found that spooling is key to obtaining DNA with adequate properties, as centrifugation and ethanol washes tend to yield much less concentrated DNA. Spooled DNA in EB buffer quickly forms a “jelly-like” mass, which is the best sign of efficient relaxation of HMW DNA. Moreover, this observed dynamic will ensure its homogenization in all subsequent steps and this DNA requires acceptable and reproducible periods of up to one day to be fully relaxed and ready for library preparation. Genomic DNA isolated by using the procedure described here showed optimal absorbance ratios and had the most reliable concentration values obtained using Qubit after light shearing. Repeated measurements were always within less than 10 % of each other. It is also worth mentioning that when isolated DNA was stored at 4°C for three months and even after freezing/thawing process, we did not observe any decrease in DNA stability compared to freshly isolated DNA.

Although this method produces extremely long DNA fragments, regarding Nanopore sequencing we find that controlled mechanical shear of HMW DNA helps to produce even longer reads. This can possibly be explained by pore blocking of entangled DNA fragments, less efficient adapter ligation but could also be genome specific. This may be particularly true for *Tribolium* genomes, which are known to be A-T rich (>60%) and highly repetitive, with individual satellite DNAs accounting for up to 40% of the genome [19]. For this reason, each species and its DNA will most likely require some degree of optimization in shearing to obtain higher N50 values. Nevertheless, this protocol succeeded in isolating HMW DNA from three closely related beetles and due to its excellent performance, especially in terms of DNA concentrations and qualities, we propose that it provides a suitable starting point for isolation of HMW DNA from other *Coleoptera* species.

## Figures and Tables

**Figure 1 genes-12-01114-f001:**
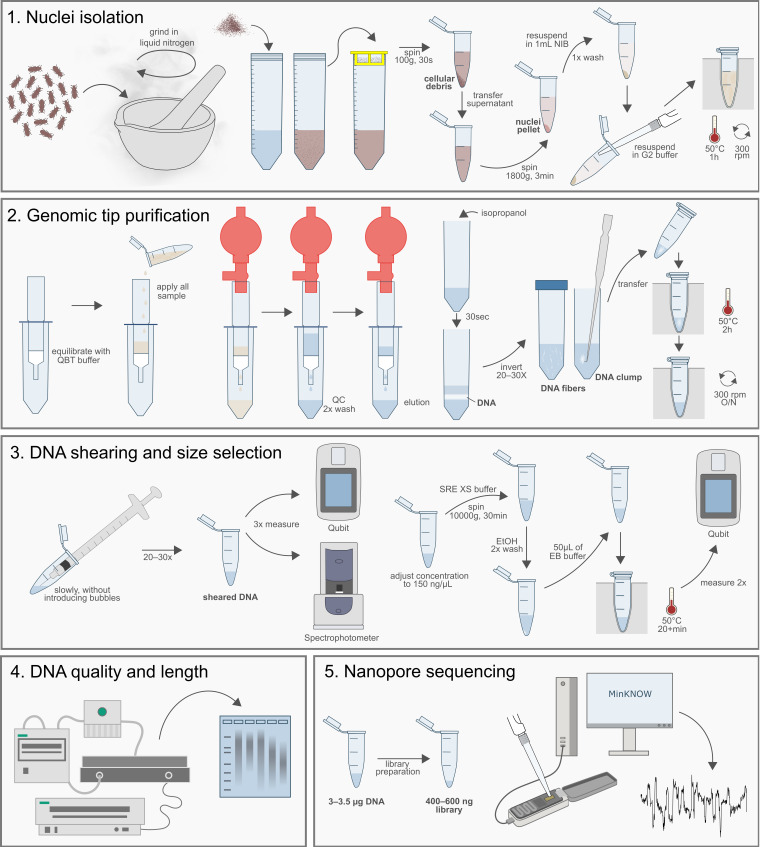
Workflow scheme of optimized HMW DNA isolation in *Tribolium* beetles, quality assessment and preparing for Nanopore sequencing. Steps are explained in detail in the Procedure subsection.

**Figure 2 genes-12-01114-f002:**
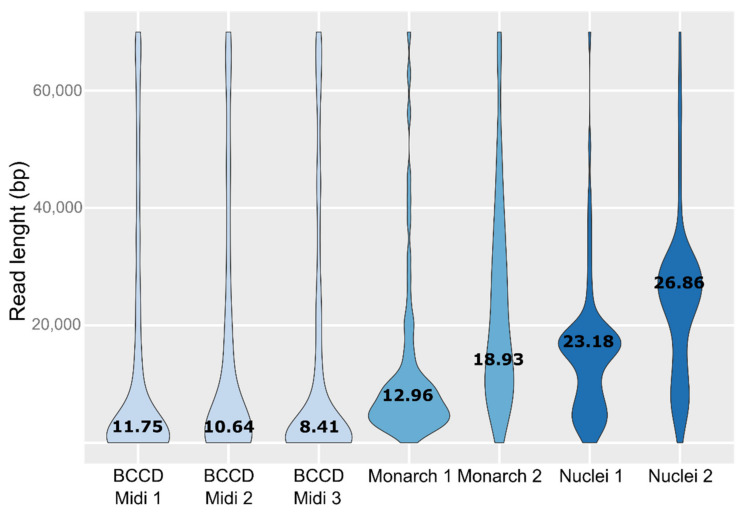
Violin plots of read length distribution of the Nanopore sequencing data derived from six different HMW isolations using Blood and Cell Culture DNA Midi kit (BCCD Midi), Monarch HMW DNA extraction kit for tissues (Monarch) and our developed protocol (Nuclei). Appended numbers represent performed experimental replicates (See also Appendix A). The width of each violin indicates the size of dataset and the number represent N50 values (kb).

**Figure 3 genes-12-01114-f003:**
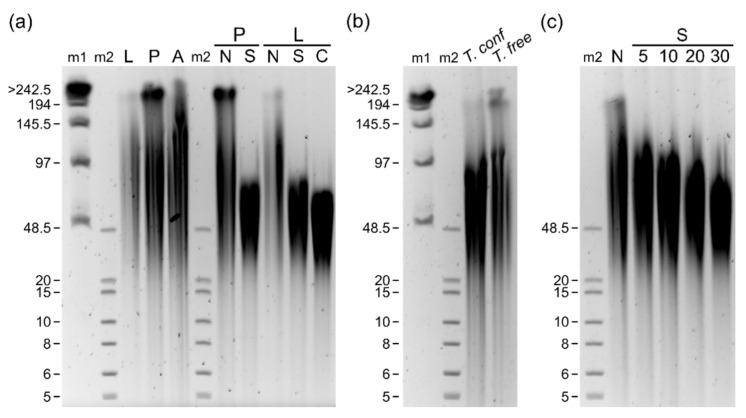
Pulsed field gel electrophoresis of isolated genomic DNA together with sheared and cleaned fractions from different developmental stages of *T. castaneum*, *T. confusum* (*T. conf*) and *T. freemani* (*T. free*) beetles. Lambda DNA (m1) and Extend DNA ladder (m2) were used as molecular weight markers. For all samples around 1 µg of DNA was mixed with loading dye and loaded per single well. Electrophoresis was run on CHEF-DR III PFGE system with 1% agarose gel with settings listed in procedure section. (**a**) Genomic DNA isolated from *T. castaneum* developmental stages (L-larvae, P-pupae, A-adults) together with sheared (30×) and size selected fraction. N-non-sheared fraction, S-sheared fraction with G30 needle, C-size selected DNA with Short Read Eliminator Kit XS (Circulomics). (**b**) Genomic DNA isolated from *T. confusum* pupae and *T. freemani* adults. (**c**) Testing of needle shears with increasing amount of passes through needle (indicated with numbers) on *T. castaneum* adult gDNA.

**Figure 4 genes-12-01114-f004:**
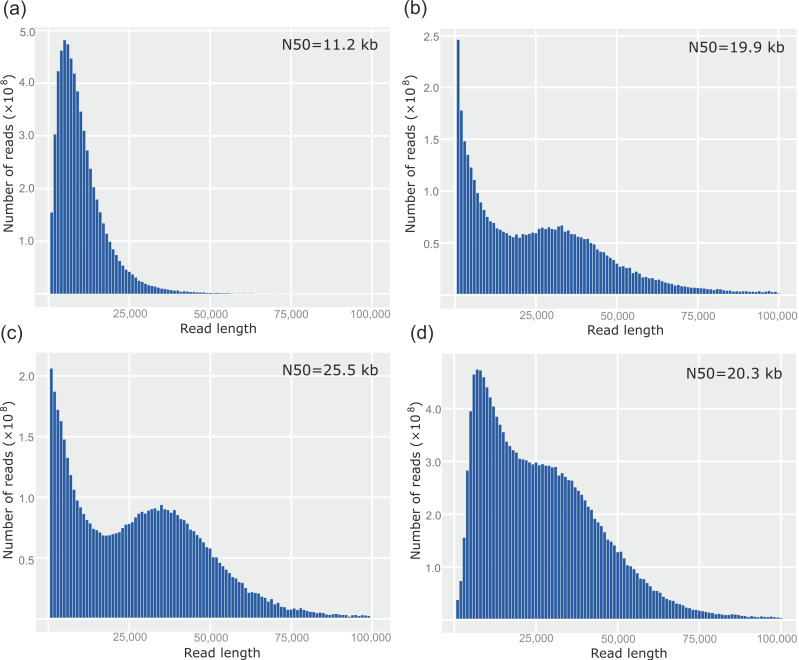
Length distribution graphs in correlation with gDNA shearing and size selection in Nanopore sequencing. Corresponding N50 values are indicated in top right corner of each graph. (**a**) DNA that has not been sheared. (**b**) Shearing with 20 passes through G30 needle. (**c**) Shearing with 30 passes through G30 needle. (**d**) Shearing with 30 passes through G30 needle and size selection with Short Read Eliminator kit (Circulomics).

**Figure 5 genes-12-01114-f005:**
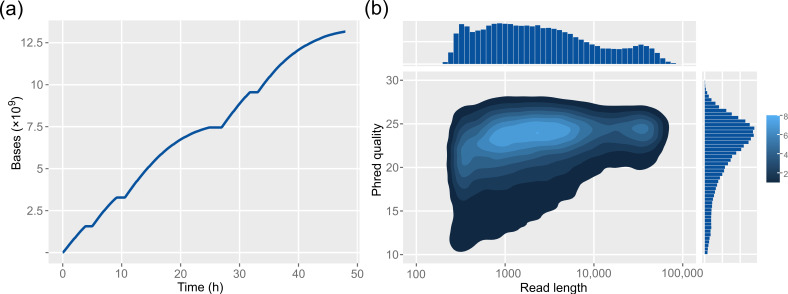
Summary of Nanopore sequencing run output. (**a**) Cumulative bases output after 48 h run on MinION flow cell with five consecutive library loads and four intermediate DNase washes in the duration of 60 min. (**b**) 2D density plot of overall read Phred quality scores and read lengths. Color shades represent higher cumulative fractions of reads with a given length and quality.

**Table 1 genes-12-01114-t001:** Overview of isolated HMW DNA samples from *T. castaneum* beetles. Amount of starting material is listed in optimal weights for successful isolation from three developmental stages (larvae, pupae, adults). Accompanying DNA concentrations, final quantities and absorbance ratios at 260 and 280 nm for quality assessment are indicated (see details in Appendix A).

	Starting Material (mg)	DNA Concentration (ng/µL)	DNA Yield (µg)	A_220/260_	A_260/280_
**Larvae**	1100	512	51.2	1.87	2.35
**Pupae**	200	172	17.2	1.87	2.25
**Adults**	1000	643	64.3	1.85	2.14

## Data Availability

No new data were created or analyzed in this study. Data sharing is not applicable to this article.

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
