# Peer review of "Isolation of High Molecular Weight DNA from the Model Beetle Tribolium for Nanopore Sequencing"

_genes, 2021, doi:10.3390/genes12081114_

Round 1

Reviewer 1 Report

Review of manuscript numer: genes-1303726 

This paper presents the method of DN isolation from model beetle Tribolium
for Nanopore sequencing. Authors applied a new steps in DNA isolation procedurÄ™ that enables obtaining good product for nanopore sequencing. This manuscript is rather procedurÄ™/methods of DNA isolation that was elaborated on small group of samples. However this paper was prepared as procedure I recommend some changes that were listed below:

  1. I propose include information about number of specimens and numer of replications used in experiments. It should be done for each stage.
  2. Procedure of isolation should be described in passive voice in past – it is typical for manuscript.
  3. From line 131 - please include detailed modifications compered to method described by Brown and Coleman (15).
  4. Line 134 – „… add the sample…” you should precise the amount of sample (numer of specimens and weight of sampe).
  5. If described method is better than other published methods you should include the hard proof of this – the best is the final molecular analysis in this study - nanopore sequencing. Did you compare DNA products, isolated by different methods, in nanopore sequencing ? If yes you should describe this in separate paragraph.
  6. I did not check the English because I am not a native-speaker.

Reviewer 2 Report

The authors provide a clear, well tested protocol comparison/development for high quality high molecular weight DNA from Tribolium. I think this work has the potential to benefit researchers interested in Nanopore sequencing, regardless of the targeted DNA. I believe the ms could be accepted as-is. I have two very minor comments, which the authors and editors may consider:

1) The English is good, but may also benefit from a final round of style editing focusing on article usage.

2) I did not understand: "In adults, due to the large amount of polysaccharide chitin in the beetle cuticle, adequate amounts of DNA were obtained only from >600 mg of starting material. The same observation was true for larvae, which have a large gut and a high 232 amount of fat."
Would "...of DNA required >600mg of starting material..." capture the intent?
